# Designing an irreversible metabolic switch for scalable induction of microbial chemical production

Ahmad A. Mannan [1] & Declan G. Bates [1]✉

Bacteria can be harnessed to synthesise high-value chemicals. A promising strategy for increasing productivity uses inducible control systems to switch metabolism from growth to chemical synthesis once a large population of cell factories are generated. However, use of expensive chemical inducers limits scalability of this approach for biotechnological applications. Switching using cheap nutrients is an appealing alternative, but their tightly regulated uptake and consumption again limits scalability. Here, using mathematical models of fatty acid uptake in *E. coli* as an exemplary case study, we unravel how the cell's native regulation and program of induction can be engineered to minimise inducer usage. We show that integrating positive feedback loops into the circuitry creates an irreversible metabolic switch, which, requiring only temporary induction, drastically reduces inducer usage. Our proposed switch should be widely applicable, irrespective of the product of interest, and brings closer the realization of scalable and sustainable microbial chemical production.

[1] Warwick Integrative Synthetic Biology Centre, School of Engineering, University of Warwick, Coventry CV4 7AL, United Kingdom. ✉email: D.Bates@warwick.ac.uk

Bacteria sense and adapt to changes through control systems encoded in their genome. Advances in synthetic biology are uncovering the principles for recoding these control systems[1–3], and enabling the creation of circuits that orchestrate the expression of genes as a non-linear or bistable on-off switch[4,5], in an oscillating pattern[6,7], as logic gates[8] and to maintain expression homeostasis[9]. A primary goal of synthetic biology is to re-wire cell metabolism for the synthesis of compounds of interest, and there is growing literature on harnessing such engineered control circuits to do that, for various biotechnological applications[10,11].

Typically, cell growth has to be sacrificed for chemical synthesis, and though one can optimise the balance between the two competing metabolic objectives, if the trade-off is severe this can still result in poor productivity[12]. A promising strategy to overcome this constraint is dynamic control. Theoretical works of[13,14] demonstrated that by allowing cells to first grow, and then inducing a switch to chemical synthesis once a large population of cell factories are generated, growth and production can be decoupled. This allows each metabolic objective to be maximised separately, and consequently enhance productivity. Recent applications of dynamic control have successfully engineered chemically responsive transcription factors (TF), such as IPTG-inducible LacI, as the switch[15,16], and have reported improved titres and volumetric productivity.

However, the need for inducers poses a problem. Use of IPTG and other gratuitous inducers are extremely expensive (e.g. £1055 for 25g IPTG, ≥99% purity, SigmaAldrich), and quickly become economically infeasible to use when scaling-up. Alternatively, quorum sensing has been repurposed to enact dynamic control, autonomously switching cells to chemical synthesis at a pre-set threshold population density[17,18]. However, the cost of autoinducer (AI) synthesis to cell fitness can delay the growth phase[19] and negatively affect productivity. Also, AI synthesis may complicate upstream processing in industrial applications, for instance, requiring close monitoring of culture state to avert delays caused by autonomous on-off switching during seed-train scale up. Finally, its applicability to different fermentation modes is unclear. Both fed-batch and continuous are economically competitive fermentation modes[20], but how culture volume increases in fed-batch or dilutions in continuous mode affect induction and its retention are not currently known. Ultimately, the goal is to achieve industrial scale production. The simplicity of manual induction is strongly attractive for industry, and current upstream processing already allows for its implementation. Maximising productivity requires optimal timing of induction[13,21], but is easily done by tracking culture density. A more fundamental problem is the high cost of the current use of gratuitous inducers. This represents a key roadblock to the scalability of dynamic control strategies, and so to economically viable industrial scale production.

We instead propose to exploit cheap natural nutrients that induce changes in gene expression. Bacteria can sense and re-orchestrate their metabolism to take advantage of nutrients when they become available[22,23], and we consider how to repurpose such control systems to implement dynamic control. Nutrients are however metabolised by the host, so their constant addition may be needed to retain the induced state, again limiting scalability. We hypothesise that re-engineering this control to create a bistable switch with binary transition of metabolism from the growth to production phenotype, irreversibly, after only temporal induction, can address this problem. Inducible bistable switches can endow a memory of the induced phenotype[4], but it is not currently known how to engineer them to behave irreversibly and retain the production phenotype even after inducer depletion.

An important class of nutrients that induce a change in gene expression are fatty acids. Long-chain fatty acids are important for maintaining cell membrane lipid homeostasis, and as such their uptake and metabolism is a universal function among a wide range of industrially relevant organisms, including E. coli[24], gram-positive bacteria (Bacillus[25], Streptomyces[26], Corynebacterium[27]), yeast and fungi (Saccharomyces[28], Rhodospuridium[29], Aspergillus[30]) and even mammalian cells[31]. Therefore, a metabolic switch inducible by fatty acids should be universally applicable. In the case of E. coli, when fatty oleic acid becomes available, a small amount is internalised via low leaky expression of transport and acyl-CoA-synthase enzymes (FadD), and on sequestering TF FadR relieves its repression on expression of the nutrient uptake and degradation enzymes, allowing further nutrient uptake in a positive feedback fashion (Fig. 1a, dashed arrow).

A number of other nutrient inducible systems share the same circuitry with positive feedback (Supplementary Table 1). Most importantly, positive feedback is a fundamental design feature that enables control systems to behave as a bistable switch in certain parameter regimes. Bistability from positive feedback has been demonstrated in natural systems, such as in signal transduction systems[32,33] and positive autoregulation with high cooperativity[34], and from synthetic genetic circuits that compose toggle switches as two mutually inhibiting TFs[4] or as transcriptional-activator-like effectors (TALEs)[35]. Nutrient inducible systems can indeed be tuned to behave as bistable switches[36], and for our application their control can be extended to regulate the expression of enzymes driving growth and production (Fig. 1a, control application). However, it is not yet known how to redesign the system and recode its functionality to switch irreversibly. Addressing this fundamental problem will allow scalable use of cheap nutrients, and overcome the need to constantly add nutrients to counter their consumption by the cell. For biotechnological applications, since this means temporary induction, using a cheap inducer, total induction costs can be drastically reduced.

In this work, we consider the oleic acid-inducible fatty acid uptake system from E. coli DH1Δ fadE strain used in[37] (Fig. 1a, cross), as an exemplary case study of a nutrient uptake system. This strain has recently been shown to have a slower reversion back to the uninduced state vs its parent DH1 strain[37], and this desirable property makes it a good starting point for engineering an irreversible switch. We developed a mathematical model of this system and used it to study how to design a bistable controller that switches irreversibly from the growth phenotype with high growth rate, to the production phenotype with low growth rate. We exhibit control designs and optimal induction regimes that minimise both total inducer used and the time to switch to the production phenotype, in the face of a reduced growth rate during induction. We find that modifying the native circuit topology so that FadR activates its own expression produces a more robust bistable switch, that requires less inducer to achieve the production phenotype and even less to sustain it. However, by augmenting the circuitry with another positive feedback loop we show that the bistable switch can behave irreversibly and drastically lower induction usage. Simulations of the induction dynamics reveal an inherent trade-off between total inducer used and time to achieve the production phenotype, but computational analysis uncovered simple design principles to improve both performance objectives.

## Results

**Designing a robust bistable-switch.** We first study how to re-engineer the fatty acid uptake system and its control on growth and production (Fig. 1a) to make it behave as a bistable switch. For this, we developed a mathematical model of the endogenous oleic acid-inducible system and add transcriptional control on the expression

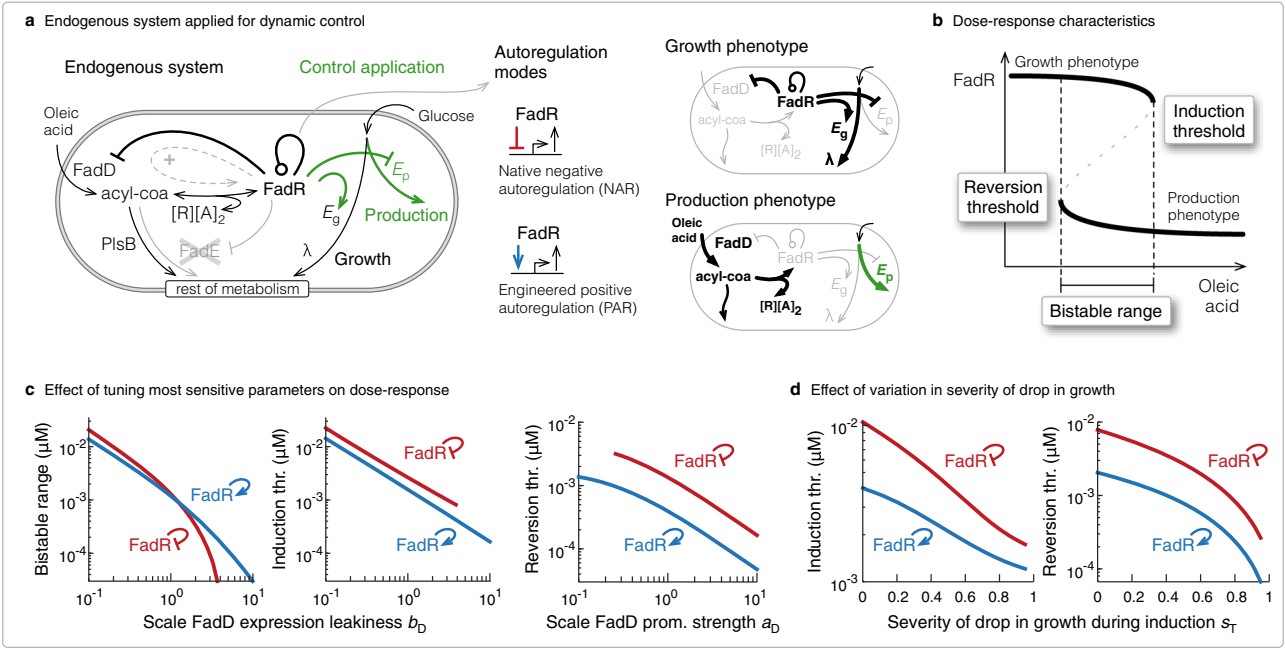

**Fig. 1 Design of a robust bistable-switch. a** Schematic of the endogenous regulation of fatty acid uptake in *E. coli* DH1Δ *fadE* strain (black text and lines), and how its regulator FadR can be applied to control an oleic acid-induced switch from growth to production (green text and lines). Those components in high abundance in the absence and presence of oleic acid, i.e., growth and production phenotypes, are highlighted in bold in illustrations on right. We consider two ways FadR regulates its own expression: the native negative autoregulation (NAR) or engineered positive autoregulation (PAR). **b** Illustration of a dose-response curve and three features we measure to characterise the switch behaviour: induction threshold; reversion threshold and bistable range. **c** Plots of how the bistable range and induction threshold are affected by changes in FadD expression leakiness, and how the reversion threshold is affected by a change in FadD promoter strength, for the native system with NAR (red) and an engineered system with PAR (blue). Parameter were varied from 10 to 1000% of their nominal value from Supplementary Table 2. **d** Plot of how induction and reversion thresholds are affected by changes to the proportional drop in growth rate after induction.

of $E_g$ and $E_p$, enzymes catalysing reactions that affect growth and product synthesis, respectively. For our modelling purposes, the specific enzymes do not matter, only that their expression can be controlled to affect growth and production, highlighting the general applicability of the proposed control system. Equations are shown in Methods section and details of their formulation and parameterisation given in Supplementary Note S2. We use this model to study how the switch behaviour, characterised by the dose-response, is affected by modifying autoregulation of sensor-regulator FadR and by tuning the circuit parameters. As defined in Methods section, this shows the state of the system, which we define as the concentration of FadR, achieved at different doses of the inducer oleic acid (OA) (Fig. 1b). However, we focus the switch characterisation on three key features of the dose-response curve: the induction threshold, reversion threshold and bistable range (Fig. 1b, see Methods section).

For engineering the system, we consider two forms of the FadR autoregulation: its native negative autoregulation (NAR), and its positive autoregulation (PAR) (Fig. 1a, right). It has recently been reported that modifying FadR with PAR slows the time it takes for the fatty acid uptake system to revert back to its uninduced state[37]. This is a desirable property, in the same spirit as switch memory, and so an important circuit topology to consider. For the system parameters, we focus on exploring only those parameters that are easily modified in the lab. It is well established that by altering the location, number and DNA sequence of an operator site for TF binding, or modifying the sequence of the RNAP binding sites, one can modify the level of leaky expression of a repressed gene, its promoter strength, and affinity the TF binds its operator upstream of the gene[3,38]. We therefore limit engineering of the system to focus on modifying the expression leakiness ($b_R$, $b_D$) and promoter strengths of TF

FadR and uptake enzyme FadD ($a_R$, $a_D$), and the affinity FadR binds its operator on its own promoter and that of *fadD* ($K_R$, $K_D$).

To achieved a bistable switch, global sensitivity analysis (see Methods section) revealed that we did not need to explore all six parameters, but could focus on two key tuning dials. Specifically, the induction threshold and bistable range are most sensitive to changes to the FadD expression leakiness ($b_D$), and the reversion threshold is most sensitive to changes to the FadD promoter strength ($a_D$) (Supplementary Fig. 2a). By varying $a_D$ and $b_D$ between 10% and 1000% of their nominal value, we discovered the circuit with PAR has a larger range of parameters where the switch is bistable, with a lower induction threshold and even lower reversion threshold, vs the native circuit with NAR (Fig. 1c, dose-responses in Supplementary Fig. 2c), i.e. it's more robust, needs less nutrient to induce the production phenotype and even less to maintain it.

In the control circuits, the cell's growth rate is also controlled. Based on the data from Usui et al.[39] (see Supplementary Note S2), we used a linear equation (Eq. (8)) to model growth rate dependence on a growth-associated enzyme $E_g$, which for example could be glucose-6-phosphate isomerase from upper glycolysis[39], whose expression is in turn activated when FadR is available (Eq. (5)). During induction, inducer oleic acid is taken up from media and converted to acyl-CoA, which in turn binds and sequesters FadR[37]. In the proposed application of the circuit, expression of growth-associated enzyme $E_g$ is no longer activated, which subsequently results in attenuated growth (Fig. 1a, production phenotype). This however affects the dilution of all other species in the system, and we wanted to understand how this feedback affects the switch behaviour. Interestingly, we found that for a greater proportional drop in growth after induction (higher $s_T$ values), the lower the induction and reversion

thresholds, irrespective of the circuit topology (Fig. 1d, dose-responses in Supplementary Fig. 2b). Though both are affected, we infer from the global sensitivity analysis that the reversion threshold is decreased most (Supplementary Fig. 2a). This in fact helps to reinforce the bistable nature of the switch, and makes it well suited to enact the dynamic control strategy, which is particularly advantageous when growth needs to be severely sacrificed for production[12].

**Tuning induction regime to sustain induced state.** We now apply the two circuits, the native with NAR and that engineered with PAR, to switch from the growth to production phenotype during the feed-batch production phase, once enough cell factories have been generated and we start to introduce OA to induce the switch. We extended the model to simulate the dynamic availability of OA (Eq. (9)) and define the induction programme in Eq. (10) (see Methods section), and used it to elucidate how to tune the induction programme and engineer the switch's parameters to address the induction performance objectives: to minimise (i) total OA used over a 100 h of the process, and (ii) the time the system first reaches the production phenotype, i.e., the switch time.

First keeping fixed the control system parameters to their nominal values, we studied how the induction performance is affected by varying the key induction tuning dial, the feed-in flux $f$ (Eq (10)), from 0.25–5 $\mu$M·h$^{-1}$. That is, we solved the multi-objective optimisation problem:

$$\min_{f} \left( J_1, J_2 \right),$$
$$J_1 = \int_{t=0}^{100} f_{in}(t) \; dt \,; \; J_2 = \min\left( t \mid R(t) \le R_{pp} \right); \quad (1)$$
$$\text{subject to } 0.25 \; \mu M \cdot h^{-1} \le f \le 5 \; \mu M \cdot h^{-1},$$

where $R_{pp}$ is the highest concentration of FadR for which the system is defined to be at the production phenotype (see Methods). This gave us a set of Pareto solutions that revealed an inherent trade-off between the two objectives: slower feed-in results in less total OA used, but at the cost of longer times to switch to the production phenotype (Fig. 2a). An optimal feed-in flux can be selected to balance the objectives and give a performance closest to the ideal point at (0,0), but interestingly the control with PAR showed a larger range of feed-in flux values where less OA is used or the switch time is less, vs circuit with NAR (Fig. 2a).

To unravel what is behind this superior performance, we selected feed-in flux $f = 1.3 \; \mu M \cdot h^{-1}$ (Fig. 2b, top plots) and simulated the induction dynamics for both circuits. Simulations revealed three fundamental differences in the dynamic response by the system with PAR vs NAR: (i) it reaches the induced state earlier (Fig. 2b, $\tau$), (ii) retains that state for longer (Fig. 2b, *) and (iii) the length of subsequent inductions needed to sustain that state are shorter (Fig. 2b, †). The key factor driving these behaviours is the lower concentration of total FadR ($R + C$) in the system after it is induced (Fig. 2b, bottom plots). During induction, almost all free FadR ($R$) are stored into an inert sequestered complex ($C$) (Fig. 2b, second and fourth bottom plots), and once induction ceases sequestration is reversed and FadR is quickly freed to again repress expression of the unneeded FadD. However, in the system with PAR, FadR's sequestration during induction impairs its self-activation, as opposed to the enhanced expression seen in the circuit with NAR (Fig. 2b, fourth vs second bottom plot). This means there is far less FadR to sequester in the system with PAR, and so consequently reaches the production phenotype earlier. Furthermore, since continued growth progressively dilutes away the pool of total FadR stored as sequestered complex, less and less FadR is freed every time induction

is ceased. Less FadR does mean shorter subsequent induction periods are needed to sustain the production phenotype, but also the resulting weaker self-activation causes the observed slower reversion (Fig. 2b, bottom far right plot).

Slower reversion is in fact an emergent property of the circuit with PAR, not simply created by a choice of parameters. We define reversion rate mathematically as the least negative eigenvalue of the system dynamics close to the uninduced steady state (growth phenotype), and in deriving the eigenvalues in terms of the system parameters (Supplementary Note S4), we found the reversion rate of the circuit with PAR is always slower than that of the circuit with NAR, irrespective of parameters (as long as they are positive valued).

To elucidate the principles of how to engineer the two switches to further reduce total OA used over the induction process, we then varied each experimentally accessible parameter at a time ($b_R$, $a_R$, $K_R$, $b_D$, $a_D$, $K_D$), from 100% to 1% of their respective nominal value in Supplementary Table 2, and recalculated the performance objectives of the Pareto solutions. We found the Pareto was almost unaffected by variations in the expression and regulation of FadD ($b_D$, $a_D$, $K_D$) (Supplementary Fig. 3), but was sensitive to changes in the expression and regulation of FadR, particularly in the circuit with FadR PAR. Specifically, a further reduction in total OA used can be achieved by an effective lowering of the amount of FadR, by reducing FadR promoter strength ($a_R$), or by decreasing self-activation ($K_R$), but an inherent limit was observed (Fig. 2a, inset plots). Intuitively, reducing FadR abundance enough will effectively break the control.

**Additional positive feedback loop enables irreversibility.** We next investigated whether the bistable-switch could be engineered to behave irreversibly, that is to retain the induced state even after inducer is completely removed. Hence, no further inductions will be required beyond an initial period, drastically reducing total induction costs over a long production process. Mathematically, a bistable switch can behave irreversibly if it can achieve more than one stable steady states when there is no inducer. Mathematical analysis, as detailed in Supplementary Note S6, proves that in fact this is not possible for either the system with NAR or PAR. A change in circuit topology is required.

For irreversibility, we need FadR to remain inactive after induction ceases. This requires a combination of two steps: (i) depleting the total pool of FadR (free FadR $R$ and complex $C$) during induction, and (ii) repressing further expression of FadR after induction ceases. We have shown that the circuit with PAR can achieve step (i), but to fulfil step (ii) we raise the hypothesis that addition of the mutual inhibition of FadR and another TF, such as TetR (Fig. 3a), can achieve this. The intuition is that during induction FadR's sequestration will allow TetR expression, which in turn will suppress further FadR expression and cause its depletion. Once induction ceases, there wont be enough FadR to repress its repressor TetR again, allowing us to retain suppression of FadR expression and so prevent reversion. However, enough of an induction period has to have been allowed to sufficiently deplete the pool of FadR.

To test this hypothesis, we developed a model of the proposed switch circuit topology illustrated in Fig. 3a. It is the same as Eqs. (5), (7) and (8), except we now model expressions of FadR ($R$) and TetR ($T$) with:

$$\frac{dR}{dt} = b_R + P(R, T) - k_f A^2 R + k_r C - \lambda R$$
$$\frac{dT}{dt} = b_T + \frac{a_T}{1 + (K_{Ri} R)^2} - \lambda T. \quad (2)$$

Here, $P(R, T)$ models the expression of FadR, which can be a function of only TetR (NoPAR, Fig. 3a, right) or both FadR and

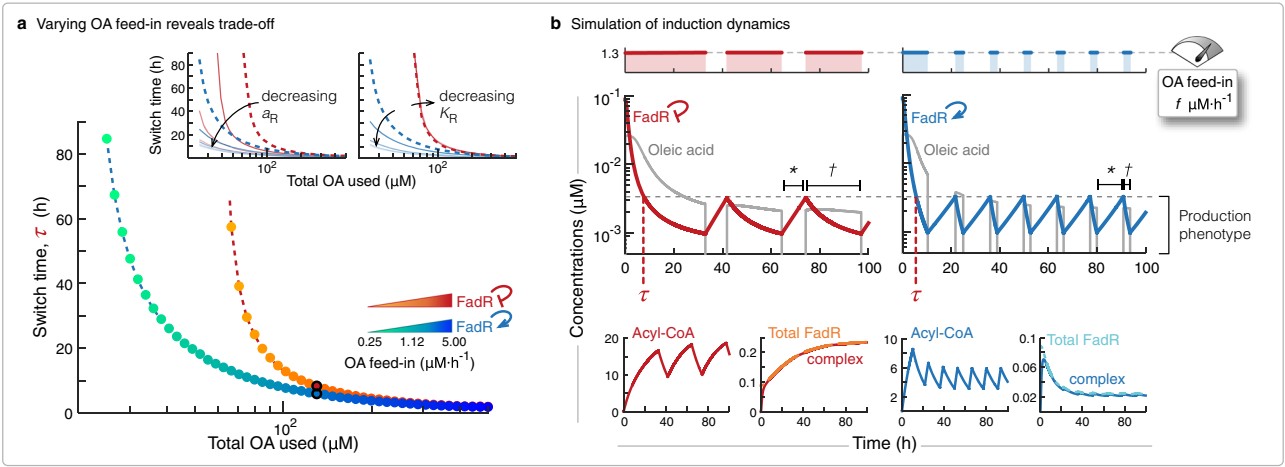

**Fig. 2 Tuning inducer feed-in and control circuit to optimise induction process performance. a** Plot of the two performance objectives for logarithmic increases in OA feed-in flux from 0.38–5 $\mu M \cdot h^{-1}$, of the circuits with NAR (yellow to red dots) and PAR (green to blue dots). Inset plots show how these curves approach a limit when decreasing control circuit FadR promoter strength $a_R$ and affinity to bind its own promoter $K_R$, from 100% (dashed lines) to 1% (lighter lines) of their nominal values in Supplementary Table 2. Red curves for system with NAR and blue curves for system with PAR. **b** Simulations of intermittent inductions of OA, at feed-in flux 1.3 $\mu M \cdot h^{-1}$ (top plots) to retain the production phenotype (middle plots), for the circuits with native NAR (left plots, red curves) and engineered PAR (right plots, blue curves). See Methods section for induction programme details. We plot availability of OA and dynamic response of FadR (middle plots), and dynamic responses of acyl-CoA, complex, and total FadR (free FadR + complex) (bottom plots). We calculate two induction process performance objectives from these simulations: total OA used (total shaded region in top plots) and switch time ($\tau$ in middle plots). Asterisk and dagger in middle plots highlights the time length of subsequent recovery and induction periods.

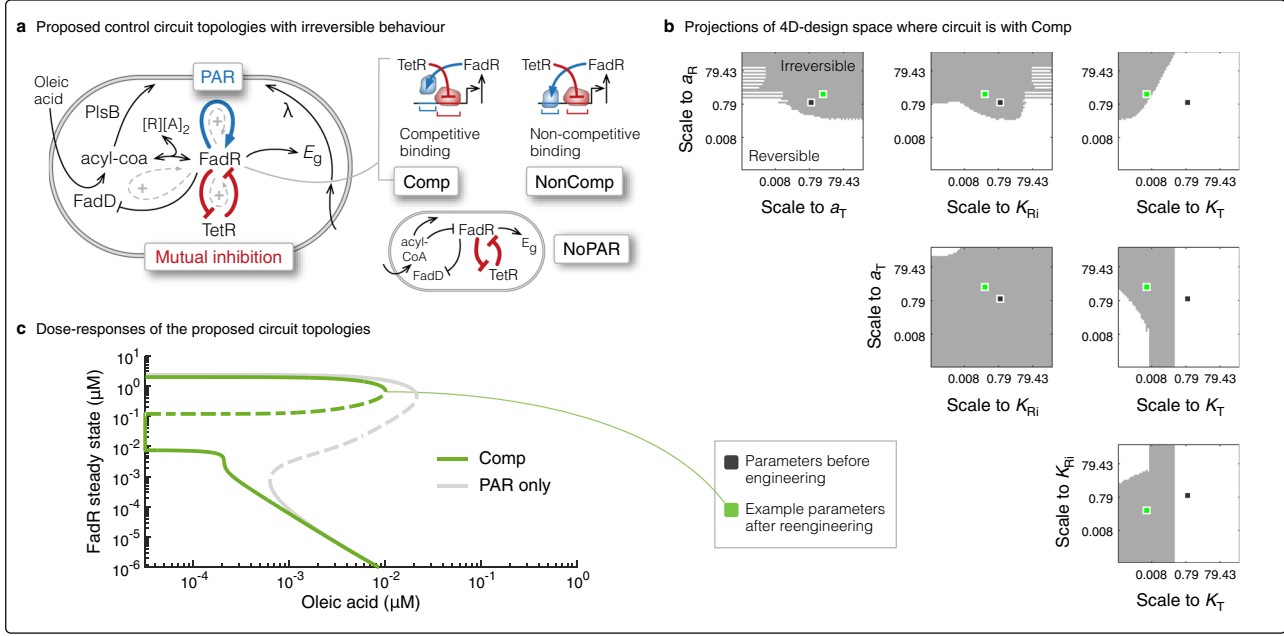

**Fig. 3 Augmenting with a positive feedback allows irreversible behaviour. a** Schematic of the control circuit with PAR (blue arrow) that was shown to deplete FadR in Fig. 2b, which we now model augmented with an addition positive feedback loop through mutual inhibition of FadR and TetR (red arrows), that we hypothesise will keep FadR expression off (see text for derivation of hypothesis). In the model of this proposed circuit, we consider three possible circuit topologies: where there is no PAR (NoPAR), and where the operator sites of FadR and TetR can be configured so both transcription factors bind competitively (Comp) or non-competitively (NonComp). **b** Plots showing projections of the parameter design space onto every pairwise plane of parameters, for the circuit Comp, the circuit we found has largest design space with irreversible behaviour of the three circuits. We varied parameters $a_R$, $a_T$, $K_{Ri}$, $K_T$ simultaneously between $10^{-3}$ and $10^3$ times their nominal values, and determined irreversible behaviour if we found more than one steady state. The dark spot indicates the nominal parameter set, which did not give an irreversible switch, and the green spot indicates an example parameter regime $((a_R, a_T, K_{Ri}, K_T) = (0.45, 0.26, 38.5, 22.3))$ that makes switch irreversible (see Supplementary Table 4 for proposed scaling to parameters). **c** Plot of the dose-response curves showing the irreversible switch behaviour of the Comp circuit (green curve), for the example parameter regime (green dot in **b**). We show comparison to the circuit with the same parameters but with only PAR, as in Fig. 1a.

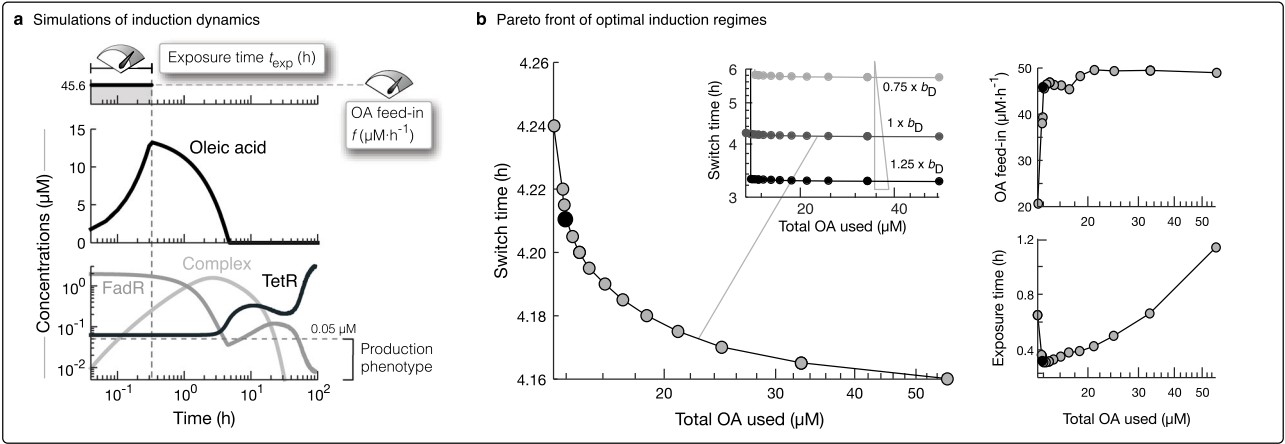

**Fig. 4 Tuning induction and irreversible switch to minimise inducer use and switch time. a** Simulation of induction with its two tuning dials OA feed-in flux ($f$) and time length of exposure ($t_{exp}$) fixed at $f = 45.6\,\mu M\cdot h^{-1}$ and $t_{exp} = 0.31\,h$ (top plot). We plot the corresponding dynamic response of oleic acid, free FadR, complex and TetR, over 100 hours of the induction process, also indicating the maximum concentration of FadR of 0.05 $\mu M$ for which we define the cell to be expressing the production phenotype. **b** Plot of the Pareto front of the induction performances achieved (left plot) for optimal feed-in flux and exposure times (right plots), as found from evaluating the multi-objective optimisation problem. The inset plot shows that increasing FadD expression leakiness by as little as 25% $b_D$ can further reduce the switch time with little affect on total OA used.

TetR. When dependent on both transcription factors, their operator sites can be arranged so their binding is competitive (Comp) or non-competitive (NonComp) (Fig. 3a, right). This gives us three possible circuit topologies to explore, and for each we derived the corresponding formula to model the FadR and TetR dependent expression of FadR as:

$$\text{NoPAR}: P(T) = \frac{a_R}{1 + (K_T T)^2},$$

$$\text{Comp}: P(R, T) = \frac{a_R K_R R}{1 + K_R R + (K_T T)^2},$$

$$\text{NonComp}: P(R, T) = \frac{a_R K_R R}{(1 + K_R R)(1 + (K_T T)^2)},$$

(3)

(see Supplementary Note S7 for derivations and parameterisation), the parameters of which are based on characterised parts in the literature, as defined in Supplementary Table 4.

To understand whether the proposed circuit topologies could behave irreversibly, we attempted to solve the model equations analytically for the steady states in terms of the parameters at OA = 0 $\mu M$. Though the equations derived (Supplementary Eq. (23)) cannot be solved analytically, we note that its solution would give us steady states of FadR in terms of primarily those parameters of the expression and regulation of FadR and TetR (see Supplementary Note S7.5): $b_R, a_R, K_R, b_T, a_T, K_{Ri}, K_T$. Interestingly, irreversibility does not depend on FadD expression ($b_D, a_D, K_D$), or the reaction kinetics of FadD ($k_{cat,D}, K_{m,D}$), PlsB ($k_{cat,B}, K_{m,B}$) or sequestration ($k_f, k_r$), highlighting that irreversibility is not dependent on how the inducing nutrient becomes available, but on the topology of the control circuit.

To uncover the parameter regimes that achieve irreversibility, we explored a four-dimensional design space of experimentally accessible parameters $a_R, a_T, K_T, K_{Ri}$, varying all parameters simultaneously between $10^{-3}$–$10^3$ times their respective nominal values from Supplementary Table 4. We found all three proposed circuits can be engineered to behave irreversibly, with the Comp circuit showing the largest parameter space in which we can design this behaviour (Fig. 3b vs Supplementary Fig. 6), i.e., the most robust design. Nevertheless, the design principles are similar between the three circuits. In particular, based on parameter values of characterised parts from the literature, we see that engineering stronger promoter strengths of both FadR and TetR ($a_R, a_T$) are needed. Also, we need to weaken the affinities of each transcription factor to inhibit the

others expression, though a stronger transcriptional inhibition of TetR expression by FadR ($K_{Ri}$) is needed vs that on FadR expression by TetR ($K_T$) (Fig. 3b). There is however a sweet-spot. If $K_T$ is weakened too far we lose the irreversibility (Fig. 3b, $a_T$-$K_T$-plane) because this effectively breaks the positive feedback in the mutual FadR-TetR inhibition and leaves us with the circuit with FadR PAR, which we know is not irreversible (Fig. 3c, grey curve).

We highlight an example parameter regime that enables all three circuits to behave as irreversible switches (Fig. 3b, c, green dot and curve), and detail in Supplementary Table 4 an example for how to engineer an irreversible switch with the Comp circuit (most robust design) in the lab. We will hereafter refer to this engineered system as the irreversible switch.

**Optimising induction and switch to minimising inducer use.** We now apply the engineered control system to irreversibly switch from growth to production during a fed-batch induction process, and revisit the question of how tuning the induction regime and parameters of the irreversible switch affect the induction performance.

We model dynamic availability of OA again by Eq. (9), but the irreversible switch behaviour now requires a different OA feed-in programme $f_{in}$. Instead of an intermittent addition based on sensing of FadR concentration, we only induce once, but now have two tuning dials: (i) the OA feed-in flux $f$, and (ii) the time length it is added for, i.e., exposure time $t_{exp}$ (Fig. 4a). We define this induction programme in Eq. (11) (see Methods section). Focusing first on exploring $f$ and $t_{exp}$ for the vector of fixed control parameters (**c**), we evaluated the following multi-objective optimisation problem to find optimal induction programmes that minimise the following induction performance objectives: OA used ($J_1$) and switch time ($J_2$):

$$\min_{\mathbf{c}, f, t_{exp}} (J_1, J_2),$$

$$J_1 = \int_{t=0}^{100} f_{in}(t)\,dt; \quad J_2 = \min(t \mid R(t) \le 0.05\,\mu M),$$

subject to $1\,\mu M\cdot h^{-1} \le f \le 50\,\mu M\cdot h^{-1}; \quad 0.1\,h \le t_{exp} \le 5\,h.$

(4)

For the switch with given parameters, we define the production phenotype when FadR $\le 0.05\,\mu M$, which can be seen on the dose-response curve in Fig. 3c (green curve).

Solving Eq. (4), we again find a Pareto front of solutions that reveal a trade-off between the two performance objectives, though interestingly, we find designs that drastically lower OA usage but make little difference to switch times (Fig. 4b). Here, reducing total OA used is achieved primarily by reducing exposure time, without needing to alter feed-in flux (Fig. 4b, right plots), though a minimum exposure time is required. These key principles enable us to set an induction programme with far superior performance, i.e., lower inducer use and switch times, than those possible with the reversible bistable-switch seen in Fig. 2a.

To elucidate how the induced state is retained after the induction period, we simulated the induction response dynamics for an example optimal induction programme, where $f = 45.6\,\mu\text{M}\cdot\text{h}^{-1}$ and $t_{\text{exp}} = 0.31$ h, (black dot in right plots of Fig. 4b). We found that during induction a high level of OA is rapidly accumulated thanks to high feed-in flux (Fig. 4a). This allows for the continuation of FadR sequestration into complex form, long after the induction period has ended. This in turn grants enough time for free FadR levels to fall and allow expression of TetR. When there is no more OA, some FadR is freed from the complex, and though it does repress TetR expression temporarily causing a dip in available TetR, it is insufficient to activate further expression of FadR and lock-down TetR expression. This results in FadR diluting away, allowing further TetR to be expressed and resulting in an effective lockdown of FadR expression.

We then looked at how tuning parameters of the irreversible switch affect performance of the induction process. We varied each of the experimentally accessible parameters in turn by ±25% of their engineered values (Supplementary Table 4), and recalculated the total OA used and switch time, for each of the optimal induction programmes of the Pareto front in Fig. 4b. We find that it is primarily the switch time that can be tuned by tuning the parameters, with little effect on total OA used (Supplementary Fig. 7). Interestingly, the switch time is most sensitive to changes in the amount of FadR and leakiness of FadD expression. In particular, tuning down FadR promoter strength ($a_{\text{R}}$), strength of PAR ($K_{\text{R}}$) or increasing FadR repression by TetR ($K_{\text{T}}$) is most effective at reducing switch times (Supplementary Fig. 7). Similarly, increasing the leaky expression of uptake enzyme FadD enables a large drop in switch time too (Fig. 4b, inset). Since reducing FadR by weakening $a_{\text{R}}$ or $K_{\text{R}}$ goes against the principles we deduced for tuning the system to achieve an irreversible switch, and tuning FadD expression does not affect that ability, we infer the key handle on switch time is therefore FadD leakiness ($b_{\text{D}}$). We infer that the key principle to reducing switch time is thus speeding the uptake of more acyl-CoA, to sequester away more FadR earlier.

There is a key design constraint though. For designs with lowest total OA used, parameter tuning can cause the optimal induction regime to no longer be sufficient to keep the system at the induced state (lines that shoot up on left side of plots in Supplementary Fig. 7). Since this problem was not as apparent for those induction programmes with slightly higher total OA used, we suggest a pragmatic approach to overcoming this problem: select a slightly suboptimal induction programme. As elucidated in the previous section, one must also beware that tuning parameters too far may make the switch behave reversibly again.

Circuitry of the irreversible switch shares a common motif with the canonical toggle switch[4]; the positive feedback loop created from mutual repression of two TFs (Fig. 3a vs Supplementary Fig. 8a). Although the fundamental advantage of the proposed irreversible switch is use of a cheaper, sustainably available nutrient inducer, rather than gratuitous inducer IPTG, and the ability to engineer it to switch irreversibly drastically reduces

oleic acid use vs the reversible switch (Figs. 4b vs 2a), it is still unclear how its performance compares to the more commonly adopted bistable toggle switch (hereafter referred to as bTS). To investigate this issue, we formulated a model of the bTS (see Supplementary Note S10 for details), applied it to switch cell phenotype from growth to production for inductions of IPTG (Supplementary Fig. 8a), and then compared its performance to the irreversible switch to elucidate key nuances and advantages of each.

As expected, the bTS can be engineered to be bistable, but interestingly we found parameter regimes where it too behaves irreversibly (Supplementary Fig. 10b). The principles of engineering irreversibility are consistent with those elucidated for the proposed irreversible switch: stronger promoter strengths of LacI and TetR but weaker affinities to inhibit the others expression (Supplementary Fig. 10b). The bTS's ability to retain the induced state after inducer washout is in line with the demonstration of the long-term stability of the induced state in Gardner et al.[4], although we predict this behaviour is conserved when applied as dynamic control.

An important difference observed in the dose-responses of the bTS (Supplementary Fig. 10a) vs proposed irreversible switch (Fig. 3c) is the higher threshold inducer concentration needed to switch to the production phenotype. In a like-for-like comparison – that is where parameters of each switch are defined such that similar levels of steady state LacI or FadR are expressed in the uninduced state and similar levels of TetR are expressed in the induced state (Supplementary Fig. 10a vs Fig. 3c) – we observe that ~0.3 μM of IPTG is needed to switch the bTS (Supplementary Fig. 10a) while only 0.01 μM of oleic acid is needed to switch the proposed irreversible switch (Fig. 3c) – approximately a 30-fold difference.

To understand how the bTS characterised by this parameter regime performs with respect to the total IPTG used and switch time, we solved a multi-objective optimisation problem to minimise both these performance objectives (defined in Supplementary Eq. (25)), and found a Pareto front of optimal IPTG feed in flux values and exposure lengths. A key difference observed in the induction performances of the bTS vs the proposed irreversible switch is the lower total IPTG used vs total oleic acid used. However, although reducing IPTG feed-in flux by ~40% can reduce total IPTG used by almost 25% (from 15.6 μM to 11.8 μM), it comes at the cost of substantially longer switch times, from 0.32 h to almost 100 h (Supplementary Fig. 11a). Conversely, for the proposed irreversible switch, we can reduce the total oleic acid used to 13.4 μM with almost no additional delay to switch times beyond 4.24 h (Fig. 4b), which can be tuned down further to a minimum ~0.39 h by increasing leaky expression of the oleic acid uptake enzyme FadD 10-fold ($b_{\text{D}}$) (Supplementary Fig. 7b). Given the small difference in switch times, we turn our attention to the total amount of inducer used. Considering the Pareto solution lying closest to the ideal point (0,0), in the case of the proposed irreversible switch we achieve total oleic acid use of 14 μM (Supplementary Fig. 7b), whereas for the bTS total IPTG use can be pushed down to ~11.8 μM (Supplementary Fig. 12). Both switches can thus be engineered with similar performances, but the significantly lower cost of oleic acid vs IPTG (£227 for 25g oleic acid vs £563 for 10g IPTG, Sigma-Aldrich at ≥99% purity) makes it a far more economically viable solution for inducing production.

## Discussion

Application of inducible dynamic control for two-stage production from microbial cell factories is highly attractive for industry.

Recent work has shown that the natural design of nutrient inducible circuitry is efficient for reverting quickly once nutrients deplete, helping to conserve cellular resources[37]. However, in an effort to reduce inducer usage for industrial biotechnology application, in this study we elucidated how to re-engineer the circuitry to counter this reversion when applying the control to switch cell phenotype from growth to production. We demonstrated that replacing the endogenous negative autoregulation (NAR) of FadR with positive autoregulation (PAR) not only created a more robust bistable switch (i.e. larger parameter space where it's bistable), but fundamentally altered the system dynamics: during induction, FadR is sequestered and can not activate its own expression, resulting in FadR, sequestered and free, to dilute away as cells grow and divide. Its resulting low abundance slowed reversion and so extended times between inductions, consequently reducing total inducer usage over a fixed process time.

Though irreversibility can reduced inducer use further, the maths predicts the circuitry with PAR cannot be engineered to behave irreversibly. FadR is eventually re-expressed and cells switch back to the growth phenotype. Therefore, to retain the production phenotype for long fed-batch or continuous fermentation (modes attractive for industrial application[20]) would require periodic additions of inducer, accumulating to a large usage. To keep repressed FadR expression during induction we augmented the circuit with a positive feedback loop via TetR, similar to the canonical toggle switch[4], and discovered this allowed it to be engineered to behave irreversibly. Consequently this drastically reduced total inducer usage vs the reversible switch with PAR. Exploration of circuit parameter easily modified in the lab elucidated key design principles to engineer irreversibility: stronger FadR and TetR promoter strengths but low affinities of each to inhibit the others expression. However, there is a sweet-spot – if repression of FadR is too weak the feedback breaks. Interestingly, arranging FadR and TetR operator sites regulating FadR expression so they bind competitively grants a more robust irreversible switch, i.e., a larger design space to engineer irreversibility.

Further mathematical analysis revealed that achieving irreversibility is in fact not dependent on any parameters associated with the expression and regulation of uptake enzyme FadD, nor on the reaction kinetics of inducer uptake, consumption or even sequestration. We therefore speculate that any other pair of nutrient and TF sequestered by that nutrient may also be constructed with the proposed circuitry and engineered with the same design principles to function as irreversible bistable switches, even in host organisms other than E. coli. For instance, many other nutrient inducible uptake systems that even share the same circuit topology as the fatty acid system studied (Supplementary Table 1), may serve as candidates for constructing alternative irreversible switches, induced by their respective nutrients. For their construction in the lab, it is important to note that two additional interventions are needed: (i) knocking-out degradation enzyme like FadE (the reason we chose to focus on E. coli DH1∆ fadE strain (Fig. 1a)) to slow inducer consumption and shorten induction exposure periods, with the added benefit of reducing protein expression burden; and (ii) engineer the promoter of uptake enzymes like FadD to allow expression without the need for activation by CRP-cAMP, to make the switch independent of growth conditions. Carbon catabolite repression is a common feature of a number of nutrient uptake enzymes in E. coli. In the case of fatty acid uptake, if E. coli is cultured on glucose, expression of FadD is inactive and little oleic acid will be taken up when available, impairing the switch function. We suggest to replace FadD promoter with a synthetic promoter with only an operator for inhibition by FadR.

There are a few caveats to bear in mind when constructing the irreversible switch circuit in the lab. Though we discovered that the switch is more robust by arranging FadR and TetR to bind competitively in their co-regulation of FadR expression, protecting its function from the deleterious effects of genetic drift poses a challenge in the long run. Harnessing the principle of growth-coupling may help. FadR's activation of $E_g$ in the circuit creates growth-coupling, and we speculate this at least creates selection pressure to preserve the circuit's control on growth. Furthermore, if the circuit controls expression of the right host-native enzymes during induction to create growth-coupled production[40], albeit with attenuated growth, this may also confer selection pressure to preserve circuit function. Another important consideration is the burden of expressing heterologous product synthesis pathway enzymes. Their expression may compete for host ribosomes and RNAP[41], which can feedback to affect circuit function[42–44]. However, since reducing cell growth rate increases the proportion of ribosome content[45], we speculate that the sacrifice of growth for production enacted by the switch may help relieve this competition. Alternatively, the use of orthogonal ribosomes to control heterologous enzyme expression has recently shown to help relieve such competition[46]. It is also important to consider how sequestering the controller TF can affect other parts of metabolism. In the case of FadR, its sequestration may affect its activation of fatty acid biosynthesis regulon[24], an essential pathway, and its repression of the glyoxylate shunt[47] in E. coli. However, in the study of Hartline et al.[37], when E. coli grown in glycerol was treated with high levels of OA there was no evidence of any effect on growth. Moreover, since the switch will decrease growth, the demand for these pathways will decrease anyway. We thus speculate that the switch will not affect those pathways significantly.

The discovery of how to engineer a nutrient inducible irreversible switch is exciting because of its universal applicability. Firstly, the use of long-chain fatty acids as inducers is arguably universal. Fatty acids are precursors to lipid synthesis and critical in maintaining membrane lipid homeostasis, so their uptake and metabolisation is a universal function amongst a wide range of organisms, including those of importance for industrial biotechnology applications such as E. coli[24], gram-positive bacteria (Bacillus[25], Streptomyces[26], Corynebacterium[27]), yeast/fungi (Saccharomyces[28], Rhodospuridium[29], Aspergillus[30]) and mammalian cells[31]. Though kinetics may vary between organisms, as explained above model analysis predicts that this does not affect the ability to engineer an irreversible switch. There is an important additional advantage in the application of a fatty acid inducible control – it has a large, ready-to-use circuit construction toolbox for Synthetic Biology. FadR has a large array of operator sites, binding to repress some promoters[48] and activate others[24], and can be sequestered by the CoA thioester derivatives of a number of long-chain fatty acids, each with different affinities, for instance oleoyl-CoA, myristoyl-CoA and palmitoyl-CoA[49]. A large library of working fatty acid inducible expression systems can thus be readily constructed, making it a versatile and widely applicable tool. As an example of out-of-box use vs the most commonly applied IPTG-inducible tac promoter, application of oleic acid inducible promoter aldA to shift E. coli growth to synthesis of GFP resulted in a twofold increase in its productivity[48].

An inducible switch commonly adopted in the lab is the canonical toggle switch[4]. Comparative analysis of its application vs the proposed OA-inducible irreversible switch to switch from growth to production predicts both switches can be engineered to achieve similar performances with respect to total inducer used and switch times. However, the substantially higher cost of IPTG vs oleic acid significantly outweighs the slightly lower total use of IPTG. With IPTG at £563 for 10 g and OA at £227 for 25 g (quoting Sigma-Aldrich for 99% purity), the predicted 11.8 μM of

IPTG used per cell would cost 15.8p while the 14 µM OA used per cell would cost 3.6p – the irreversible switch enabling a substantial saving of over 77%.

Application of the irreversible switch can also greatly simplify implementation of induction – only a predetermined, short induction period is required, there is no need to continuously sense the system state to decide when to add or cease induction. This does leave us with two tuning dials to implement induction, but interestingly, we discovered that reducing time length of inducer addition can reduce total inducer use, and increasing FadD expression leakiness, to allow greater inducer uptake and earlier FadR sequestration, reduced the switch time. Moreover, the effect of tuning each dial on the other objective was negligible. However, we found that reducing switch time to far then requires slightly longer periods of inducer addition to ensure FadR is sufficiently depleted and enough TetR is expressed to keep FadR expression locked-down.

The use of metabolised inducers such as oleic acid presents another key advantage – their rapid depletion from the media eases purification of the product in downstream processing. Downstream processing is a significant share of the total operating and manufacturing costs of fermentation products[50], and the need to remove remaining inducer in media may present additional costs. Our simulations predict that additions of non-metabolisable gratuitous inducer IPTG lingers in the media well after temporal induction has ceased (Supplementary Fig. 11b), whereas oleic acid is predicted to be completely consumed after induction (Fig. 4a), eliminating the associated costs of removing it in downstream processing.

The proposed nutrient-inducible irreversible switch has important implications for synthetic biology, and can contribute strongly to efforts in metabolic engineering and industrial biotechnology, as a genetically encoded tool to enact dynamic control, bringing closer the realisation of scalable and sustainable production of high-value chemicals from microbes.

## Methods

**Model of oleic acid-induced switch from growth to production**. We developed a model of the dynamics of the oleic acid-inducible control system from E. coli as a system of ODEs. Full details of its formulation and parameterisation can be found in Supplementary Note S2, but we outline the model equations here. The model is of two parts: (i) the native fatty acid uptake system, from E. coli DH1Δ fadE strain (Fig. 1a, endogenous system), and (ii) the control of oleic acid-responsive transcription factor (TF) FadR on the expression of enzymes directly affecting growth and production (Fig. 1a, green arrows).

The first part of the model is adopted from Hartline et al.[37], and models the expression of TF FadR ($R$) and uptake enzyme FadD ($D$), and their control as a function of FadR, and the dynamic availability of fatty acyl-CoA ($A$) and acyl-CoA-sequestered FadR complex ($C$) as follows:

$$
\begin{aligned}
\frac{dR}{dt} &= b_R + P_R(R) - k_f A^2 R + k_r C - \lambda R, \\
\frac{dD}{dt} &= b_D + \frac{a_D}{1 + (K_D R)^2} - \lambda D, \\
\frac{dA}{dt} &= \frac{k_{cat,D} \cdot OA}{K_{m,D} + OA} \cdot D - \frac{k_{cat,B} \cdot A}{K_{m,B} + A} \cdot B - 2 \cdot (k_f A^2 R - k_r C) - \lambda A, \\
\frac{dC}{dt} &= k_f A^2 R - k_r C - \lambda C,
\end{aligned}
\tag{5}
$$

where $P_R(R)$ models the autoregulation of FadR. We consider two configurations of FadR autoregulation: (i) its negative autoregulation (NAR), as is in the native circuitry, and (ii) its positive autoregulation (PAR), as engineered in Hartline et al.[37] (Fig. 1a, right), and model them as:

$$
\begin{aligned}
\text{NAR} &: P_R(R) = \frac{a_R}{1 + K_R R}, \\
\text{PAR} &: P_R(R) = \frac{a_R K_R R}{1 + K_R R}.
\end{aligned}
\tag{6}
$$

The second part of the model captures control of FadR on growth. We first model the FadR-activated expression of a growth-associated enzyme $E_g$ as:

$$
\frac{dE_g}{dt} = \frac{a_g K_g R}{1 + K_g R} - \lambda(E_g) \cdot E_g,
\tag{7}
$$

and then used a linear equation to phenomenologically model growth rate as a function of $E_g$:

$$
\lambda = \lambda_{max} \cdot \left(1 - s_T \cdot (1 - E_g)\right),
\tag{8}
$$

where $s_T = 1 - \frac{\lambda_{min}}{\lambda_{max}}$ represents the severity of growth attenuation or the proportional drop in growth rate when $E_g$ is no longer expressed, and $\lambda_{min}$, $\lambda_{max}$ represent growth rates at zero and max expression of $E_g$. This phenomenological model of growth was derived using data from the experiments of Usui et al.[39], who showed that for successive decreases in the expression of upper glycolysis enzyme glucose-6-phosphate isomerase (Pgi) growth rate decreased linearly (see Supplementary Note S2).

To keep this model independent of the specific product of interest, and a minimal description of the system, we did not model the expression and control of a product synthesis enzyme $E_p$. Its expression is exclusively dependent on FadR and does not feedback onto the system. We instead define the production phenotype by a low concentration of FadR, $R = 0.0033$ µM. This assumes the FadR operator upstream of gene encoding $E_p$ is engineered to be that upstream of fadD, and represents the FadR concentration threshold that achieves the concentration of the respective gene product that is halfway between its min and max.

**The dose-response to characterise switch behaviour**. We characterise the switch behaviour from a plot of the dose-response curve (Fig. 1b), generated by solving for the steady states of FadR at logarithmically spaced set values of inducer OA. We determined them numerically, first deriving a nullcline in terms of FadR and model parameters for each of the two main models (see Supplementary Note S2.4 for the reversible switch model and Supplementary Note S7.4 for the irreversible switch model), then using bisection as a method to determine the steady states at its roots. To determine their stability nature, we substituted the steady state values into the analytically derived Jacobian matrix, and solved for its eigenvalues. If all eigenvalues have a negative real part, with at least one eigenvalue with a non-zero real part, then we defined that steady state stable (solid part of dose-response curve), else if at least one eigenvalue had a positive non-zero real part then we defined that steady state unstable (dashed part of dose-response curve).

We used the dose-response to distinguish between switch designs, and measured three key features of the curves: (i) the induction threshold, the concentration of OA needed to switch the system to the production phenotype; (ii) the reversion threshold, the highest concentration of OA needed to revert back to the growth phenotype; and (iii) the bistable range, the range of OA concentration where the system can be in one of two possible phenotypic state (Fig. 1b).

**Global sensitivity analysis**. To understand how the system parameters of the endogenous system in Fig. 1a affect the dose-response curve, we first looked to determining the parameters for which the induction threshold, reversion threshold and bistable region are most sensitive. To determine those parameters we applied the method of the extended Fourier Amplitude Sensitivity Test, or eFAST, using the MATLAB toolbox published in Marino et al.[51]. The method generated 100 random sets of the experimentally accessible parameters $b_R$, $a_R$, $K_R$, $b_D$, $a_D$ and $K_D$ between 10% and 500% of their respective nominal values in Supplementary Table 2, and the proportional drop in growth rate parameter $s_T$ between 0.1 and 0.9, and calculated the first-order sensitivity indices. Due to the random nature of the sampling, we performed $n = 3$ replicates and calculated the average and standard deviation of the sensitivity indices for each parameter (Supplementary Fig. 2).

**Simulating induction dynamics during production phase**. To simulate induction dynamics, we incorporated the dynamic availability of OA with the addition of ODE:

$$
\frac{dOA}{dt} = f_{in} - \frac{k_{cat,D} \cdot OA}{K_{m,D} + OA} \cdot D.
\tag{9}
$$

The second term refers to the specific consumption rate of inducer, and $f_{in}$ states OA feed in regime, which we now define. For the reversible switch, the induction process involves a FadR-dependent intermittent feed of OA to retain the production phenotype. We feed in OA at a fixed flux of $f$µM·h⁻¹ and cease it once enough FadR is sequestered that its concentration falls well below the concentration that qualifies the cell being in the production phenotype, which we define as FadR $\leq 0.3 \times 0.0033$ µM. It is important to note that the precise scaling factor of 0.3 does not change the results, as long as it is <1. When induction is ceased, the reversible nature of the switch means the system starts to revert back to the growth phenotype. To sustain the production phenotype, further OA is intermittently added at $f$µM·h⁻¹, and we define this induction programme:

$$
f_{in}(R) = \begin{cases} f \in \mathbb{R}^+, & \text{for } R > 0.0033 \ \mu M \cdot h^{-1}, \\ 0, & \text{for } R \leq 0.3 \times 0.0033 \ \mu M \cdot h^{-1}. \end{cases}
\tag{10}
$$

In the case of the irreversible switch, only a single initial period of OA feed-in is required at flux $f$µM·h⁻¹, and for time length, or exposure time period $t_{exp}$. We

define this induction regime as:

$$f_{in}(t_{exp}) = \begin{cases} f \in \mathbb{R}^+, & \text{for } t \leq t_{exp}, \\ 0, & \text{for } t > t_{exp}. \end{cases} \qquad (11)$$

**Multi-objective optimisation**. In implementing the induction process, we explored how to tune induction and the control circuit parameters, within bounded intervals of each of their values, with the aim of minimising two key induction process performance objectives: (i) total OA used over a 100 h of the induction process (Eq. (10)), and (ii) first time the system reaches the production phenotype, or switch time. We posed this as a multi-objective optimisation problem, formally stated in Eq. (1) for the reversible switch, and Eq. (4) for the irreversible switch, and used the gamultiobj function from the Global Optimization Toolbox in MATLAB 2018a to solve it, and find a set of Pareto solutions of the optimal objective values and the corresponding parameter sets.

**Reporting summary**. Further information on research design is available in the Nature Research Reporting Summary linked to this article.

## Data availability
Example nutrient inducible systems reported in Supplementary Table 1 were sourced from EcoCyc[47]. All relevant data generated and analysed in this study, and data used from referenced published sources used for model parameterisation, are included within the paper and its Supplementary Information, and are also available at GitHub https://github.com/AhmadMannan/IrreversibleMetabolicSwitch[52].

## Code availability
All computational models and analyses were conducted in MATLAB R2018a. The authors declare that all MATLAB scripts of the models of the reversible, irreversible and bistable toggle switches, and code performing the analyses replicating the results and plots presented in the paper and Supplementary Notes, are available at GitHub https://github.com/AhmadMannan/IrreversibleMetabolicSwitch. The code repository is deposited in Zenodo and citable with https://doi.org/10.5281/zenodo.474066[52]. Note that raw data points that were not reported in the published source, for instance for model parameterisation, were regenerated from a snapshot of the data plotted in the source paper using web based application WebPlotDigitizer[53].

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

## Acknowledgements
This work was funded by BBSRC Research Grant BB/M017982/1. We would like to thank Christophe Corre of University of Warwick, UK, and Yoshihiro Toya of Osaka University, Japan, for helpful discussions.

## Author contributions
A.A.M. and D.G.B. designed the research, analysed the results and wrote the manuscript. A.A.M. developed the models, performed mathematical analyses and constructed the figures.

## Competing interests
The authors declare no competing interests.
