## [Peer Review File · Nature Communications]

Reviewers' Comments:

Reviewer #1:

Remarks to the Author:

The paper describes a different approach for dynamic control of metabolism. Specifically the paper introduces an irreversible bistable switch through the use of a positive feedback that is well known to introduce bistability. The authors perform a purely computational analysis of two new switch architectures and show that their new architecture leads to bistability. Their work is thorough on the technical side though their explanation of the work is a bit patchy. Nevertheless this reviewer enjoyed reading this paper though there are several areas which require significant improvement.

Major Revisions

Authors don't do enough justification to the knowledge around bi stability. Perhaps they should cite Becskei & Serrano 2001 paper in EMBO J that really analyzes this quantitatively.

In theory, a toggle switch which is reversible and can be set-off using a QS would also work as well. Perhaps the authors can better motivate why an irreversible switch would find more application ?

Authors should also modify Figure 1a so as to more explicitly show the element of the genetic circuit with the input metabolite, output metabolic pathway and the two different positive feedback elements (that is the FadR auto regulation and the "another positive FB loop described in 80 (may be elaborate)

Again Figure 3a the circuit is not described well. But it seems like the authors have added in a new TF TetR that inhibits FadR which seems very similar to a Toggle switch that can cause bistability. I feel that the authors have to distinguish their work from the TS in terms of burden which I thought was an advantage but I am not sure.... Also the tS seems more universal than this circuit that is specific for the FA pathway.

Authors also need to include a discussion where they can compare and contrast this circuit with a TS based bistable switch in terms of speed, metabolic burden etc...

Minor Comments

Line 120 is not clear. What happens during induction ? What is being induced ? What is the off state and on state ? Please depict this better in Figure 1a

Reviewer #2:

Remarks to the Author:

Designing an irreversible metabolic switch for scalable induction of microbial chemical production
Mannan and Bates

The manuscript by Mannan and Bates tackles the well-known challenge of switching microbial factories from growth to production phase in a manner optimal for production. Some solutions to this problem use inducer molecules, coupled with bistable switches, to ensure switching. Inducer molecules are generally too expensive to be industrially relevant so the authors describe a way to harness cheap metabolites combined with irreversible switching behaviour. Their design could be economically feasible and have an impact on industrial biotechnology.

The paper is well written and the figures easy to follow. The methods and model descriptions are thorough, and the analysis very clearly described. I have some comments and suggestions for improvement.

COMMENTS

The authors claim their approach can drastically reduce induction costs. However, while this may be the case, they haven't mentioned anything about the downstream process of purification and removal of inducer molecules from product. They should comment on this, particularly the relative cost of removing nutrient oleic acid vs standard inducers.

Ln 41-42: The authors claim that quorum sensing systems are not suitable for autonomous induction because "after switching, autoinducer may not be synthesised". However, this is not a general property of using quorum sensing molecules and may only be true for certain designs for switching.

Ln 291: The authors claim "the proposed control design drastically cuts induction costs". But has this actually been demonstrated unequivocally? Since this is one of the main claims of the manuscript, I think some comparison with existing proposed designs is required.

TYPOGRAPHICAL

Ln226: Fig.4(c) -> Fig. 3(c)

Ln229: Fig.4(b) and (c) -> Fig. 3(b) and (c)

Ln233: feed-batch -> fed-batch

Response to Reviewers

Designing an irreversible metabolic switch for scalable induction of microbial chemical production
Ahmad A. Mannan and Declan G. Bates

We thank the reviewers for their very positive feedback and greatly appreciate their comments and suggestions to help improve the manuscript. Please find below a point-by-point response to each point raised by each reviewer, and we have incorporated additional (red) text on each point in the manuscript. Most importantly, we have added further work where we have developed a model of the canonical genetic toggle-switch applied to control growth and production, and performed analysis of its steady state and dynamics for comparison with the proposed irreversible switch – please see details in Supplementary Note S10.

Reviewer #1

The paper describes a different approach for dynamic control of metabolism. Specifically the paper introduces an irreversible bistable switch through the use of a positive feedback that is well known to introduce bistability. The authors perform a purely computational analysis of two new switch architectures and show that their new architecture leads to bistability. Their work is thorough on the technical side though their explanation of the work is a bit patchy. Nevertheless this reviewer enjoyed reading this paper though there are several areas which require significant improvement.

Thank for your positive comments and we are glad to hear you have enjoyed reading this paper.

Major Revisions

1. Authors don't do enough justification to the knowledge around bi stability. Perhaps they should cite Becskei & Serrano 2001 paper in EMBO J that really analyzes this quantitatively.
 - Thank you for this important point, and we apologise for the omission. We have now reworked the introduction to give greater justification to the knowledge around bistability. Specifically, we highlight studies of natural and synthetically engineered inducible systems with positive feedback circuitry that were demonstrated to behave as bistable switches, including the suggested study of Becskei & Serrano 2001. Please see lines 74-78.
2. In theory, a toggle switch which is reversible and can be set-off using a QS would also work as well. Perhaps the authors can better motivate why an irreversible switch would find more application?
 - Thank you for raising this important point. We have reworked the introduction to expand our discussion on the challenges of using QS-based dynamic control in an industrial setting, the attraction of the simplicity and general applicability of manual induction, and the need to engineer an irreversible switch for applying a metabolised nutrient for induction.
 - Specifically, we first highlighted three important points related to the difficulty of using QS-based switch in current industrial settings:
 - o *“However, the cost of autoinducer (AI) synthesis to cell fitness can delay the growth phase (Ruparell2016) and negatively affect productivity. Also, AI synthesis may complicate upstream processing in industrial applications, for instance, requiring close monitoring of culture state to avert delays caused by autonomous on-off switching during seed-train scale up. Finally, it’s applicability to different fermentation modes is unclear. Both fed-batch and continuous are economically competitive fermentation modes (Yang2019), but how culture volume increases in*

fed-batch or dilutions in continuous mode affect induction and its retention are not currently known.” – lines 40-47.

- We then re-emphasise the attraction, general applicability and ease of implementing manual induction in current industrial fermentation settings:
 - o *“Ultimately, the goal is to achieve industrial scale production. The simplicity of manual induction is strongly attractive for industry, and current upstream processing already allows for its implementation.” – lines 46-48.*
 - Finally, we focus on the fundamental problem of using gratuitous inducers. We discuss the attraction of inducing with far cheaper, sustainably sourced nutrients, but clarify that their metabolization means we need to engineer an *irreversible* switch to retain the induced phenotype after inducer depletion. See lines 53-61.
3. Authors should also modify Figure 1a so as to more explicitly show the element of the genetic circuit with the input metabolite, output metabolic pathway and the two different positive feedback elements (that is the FadR auto regulation and the "another positive FB loop described in 80 (may be elaborate)
- Thank you for this suggestion. The purpose of Fig.1a is to firstly illustrate the endogenous oleic acid inducible control system from *E. coli* and its application to control the expression of enzymes that drive growth and chemical synthesis, which we denoted as E_g and E_p . It does not matter what these enzymes are, only that they can be exploited to control growth and production, highlighting the general applicability of the circuit for production of numerous compounds of interest. We emphasize this important point in lines 104-107.
 - Illustration of the additional feedback loops that enable the circuit to be engineered to behave irreversibly is shown later in Fig.3a, in line with the presentation of the results. Line 96 (previously line 80) in the introduction points to the results to come.
 - We were not quite clear what the reviewer meant by “show the element of the genetic circuit with input metabolite, output metabolic pathways ...”. However, we have expanded Fig.1a to include illustrations that highlight the interactions active in the absence and presence of inducer oleic acid, i.e. the growth and production phenotypes, respectively.
4. Again Figure 3a the circuit is not described well. But it seems like the authors have added in a new TF TetR that inhibits FadR which seems very similar to a Toggle switch that can cause bistability.
- The reviewer is correct in their interpretation, and we apologise for any confusion.
 - In brief, our analysis of simulation dynamics in the second subsection of the results showed that reversion was driven by the eventual re-expression of FadR. From this we deduced that inhibition of FadR during induction, which led us to add a toggle-switch-like motif, may keep its expression off and achieve an irreversible switch – this became our working hypothesis. This discussion is detailed in lines 209-217 (second paragraph of the third subsection of results) and reiterated in lines 369-370.
 - We appreciate that Fig.3a may have lacked clarity due to the use of the abbreviated names of species to reflect the variables of the model. We have therefore changed the figure: replacing abbreviations of each species with their full names, similar to Fig.1a, and adjusted the caption to better explain that the proposed circuit was derived based on results from the second subsection of results, so illustrating our hypothesis (please see Fig.3 caption).
5. I feel that the authors have to distinguish their work from the TS in terms of burden which I thought was an advantage but I am not sure....
- Thank you for this suggestion. We assume the referee uses the term burden in reference to ribosome competition when over-expressing proteins.

- We agree with the intuition of the reviewer that the proposed irreversible switch may confer some advantage over the canonical TS in terms of burden:
 - In the absence of inducer, both TS and irreversible switches are expressing two enzymes: growth associated enzyme (E_g) and the inducer responsive TF (LacI or FadR); and when induced, both switches again express a net of two other enzymes, TetR and the product synthesis enzymes (E_p).
 - However, unlike the passive diffusion of IPTG for the TS, the irreversible switch requires active transport of fatty acids and their conversion to acyl-CoA thioesters (that sequester FadR). In *E. coli* these are catalysed by FadL and FadD, so two further enzymes are required to be expressed during induction, though the current circuitry can be modified to turn them back off at a high threshold concentration of TetR once the induced state is achieved.
 - Interestingly, though the irreversible switch can present a temporal additional protein expression (burden) in the host during induction, this may be overcompensated by the suggested deletion of the unrequired enzymes of fatty acid degradation (such as FadE in *E. coli*, as shown in Fig.1a and suggested in lines 386-388) and beta-oxidation (which are at least 4 additional enzymes). Since fatty acid metabolism is universal across many industrially relevant strains (lines 417-422), we speculate this strategy may reduce overall cell burden by relieving competition of the enzyme fraction of the total protein pool in the growth-poor-induced production state. We now mention this in line 388.
- It seems to us that this point overlaps with point 7 below. Please see our response there for discussion of the comparisons between the canonical toggle switch and irreversible switch.

6. Also the tS seems more universal than this circuit that is specific for the FA pathway.

- Thank you for raising this important point. We see the irreversible switch as being just as universal as the toggle switch for a number of reasons:
 - Fatty acids can be used as an inducer in many organisms. They are the precursors of lipid synthesis and critical in maintaining membrane lipid homeostasis, so their uptake and degradation is a universal function across many organisms, including those of industrial importance such as *E. coli*¹, *Bacillus subtilis*², *Streptomyces*³, *Corynebacterium*⁴, *Saccharomyces*⁵, *Rhodospiridium*⁶, *Aspergillus*⁷, and even mammalian cells⁸. – this point is emphasised now in lines 63-67 and again in lines 416-422.
 - For wider application, fatty acid inducible FadR may be even more versatile than a LacI based controller like the TS. There is a large number of ready-to-use operators (based on the large cluster of fad and fab genes controlled by FadR in *E. coli*¹) and a number of different fatty acids can be used to sequester FadR with different affinities, such as myristoyl-CoA and palmitoyl-CoA⁹ for instance. This is compared to LacI of the bistable TS which is inducible with IPTG or TMG. As demonstrated in Mannan et al¹⁰, this grants flexibility in shaping the dose-response of fatty acid inducible expression. – we have added this discussion in lines 423-431.
 - Also, though the regulation and kinetics of fatty acid uptake and degradation may vary between different selected host organisms, our modelling work predicts those parameters do not affect the ability to create an irreversible switch (see S7.5 and main text lines 230-237). We therefore infer that constructing an irreversible switch using the principles elucidated in our study can be achieved in other host organisms, and even with other nutrients-TF pairs, such as those listed in Table S1. – We added this discussion in lines 380-385. Just as Gardner et al (2000) elucidated the principles by which to design a toggle switch from any two mutually antagonising and small molecule responsive transcription factors (LacI and TetR), so too our work elucidates

the circuit topology and principles for how to tune the interactions between two mutually antagonising TFs, one of which is sequestered by a metabolised nutrient, to achieve an irreversible switch, regardless of which specific TFs are used.

7. Authors also need to include a discussion where they can compare and contrast this circuit with a TS based bistable switch in terms of speed, metabolic burden etc...
 - We thank the reviewer for this important suggestion. We have therefore performed further work to analyse a TS applied to control growth and production, performed the same mathematical analysis as that done for the irreversible switch (details of model and results in Supplementary Note S10), and performed detailed comparative analysis with the proposed irreversible switch.
 - S10 is composed of (i) the model formulation and parameterisation of the applied canonical TS, with the same circuit motif as in ¹¹; (ii) steady state analysis to determine parameter regimes where the TS can behave irreversibly; and (iii) analysis of induction dynamics and optimization of induction regime to minimise both total IPTG usage and switch time.
 - Please see the new additional analysis reported in the final subsection of results, lines 310-351 for a comparison of the proposed irreversible switch with the bistable TS.
 - We also added a further discussion of the implications of this comparative analysis in the discussion section, lines 433-439.

Minor Comments

8. Line 120 is not clear. What happens during induction ? What is being induced ? What is the off state and on state ? Please depict this better in Figure 1a
 - Thanks for bringing this to our attention. We have now adjusted the text (lines 139-142) to describe more explicitly what happens during induction with oleic acid, and have adjusted Fig.1a and its caption (additional text in red) to illustrate the circuit components that are in high abundance with and without oleic acid, i.e. the off and on states, respectively.

Reviewer #2

The manuscript by Mannan and Bates tackles the well-known challenge of switching microbial factories from growth to production phase in a manner optimal for production. Some solutions to this problem use inducer molecules, coupled with bistable switches, to ensure switching. Inducer molecules are generally too expensive to be industrially relevant so the authors describe a way to harness cheap metabolites combined with irreversible switching behaviour. Their design could be economically feasible and have an impact on industrial biotechnology.

The paper is well written and the figures easy to follow. The methods and model descriptions are thorough, and the analysis very clearly described. I have some comments and suggestions for improvement.

We thank the reviewer for their very positive comments, and for their suggestions to help improve the manuscript.

COMMENTS

1. The authors claim their approach can drastically reduce induction costs. However, while this may be the case, they haven't mentioned anything about the downstream process of purification and removal of inducer molecules from product. They should comment on this, particularly the relative cost of removing nutrient oleic acid vs standard inducers.
 - Thank you for raising this interesting point and we apologise for the omission.

- We have added an additional study to help compare and contrast the proposed irreversible switch to the more commonly applied IPTG-inducible canonical toggle switch, for dynamic control (details of the modelling work and computational analyses can be found in Supplementary Note S10).
 - We have added a discussion of the result that our models predict that use of metabolizable inducer such as oleic acid gives the advantage that it is completely consumed relatively shortly after inducer addition ceases, whereas a non-metabolizable inducer like IPTG continues to linger, and so additional costs may be incurred to separate it from product in downstream processing – please see lines 450-456.
2. Ln 41-42: The authors claim that quorum sensing systems are not suitable for autonomous induction because “after switching, autoinducer may not be synthesised”. However, this is not a general property of using quorum sensing molecules and may only be true for certain designs for switching.
- Thank you for raising this important point. We do acknowledge recent work that has demonstrated the application of quorum sensing for autonomous dynamic control in the lab (lines 39-40). Though this is an attractive strategy, there are a number of open questions regarding its general applicability in industrial settings.
 - We have thus reworked the introduction to better detail the challenges involved in applying QS-based dynamic control in an industrial setting, specifically: (i) delays in growth caused by the cost to cell fitness of autoinducer synthesis may elongate the growth phase and reduce productivity; (ii) complications in upstream processing, such as close monitoring to avert autonomous on-off switching during seed-train scale up; and (iii) uncertainty as to whether an induced state can be retained for long-term fermentation when applying QS control in fed-batch and continuous culture. Conversely, the simplicity and general applicability of manual induction is attractive to industry as it is easily implemented without changing upstream processing steps. – Please see lines 41-48.
3. Ln 291: The authors claim “the proposed control design drastically cuts induction costs”. But has this actually been demonstrated unequivocally? Since this is one of the main claims of the manuscript, I think some comparison with existing proposed designs is required.
- We thank the reviewer for this important suggestion. We have therefore added an additional theoretical model and study where we have evaluated the performance of the commonly applied canonical toggle switch to control the switch from growth to production. The details of the model formulation, steady state analysis and optimisation of the induction program, to minimise total inducer use and switch time is given in Supplementary Note S10.
 - In brief, our evaluation found that both the IPTG-inducible toggle switch and oleic acid-inducible irreversible switch proposed can be engineered to achieve similar performance, but the substantially lower cost of oleic acid vs IPTG predicted a 77% reduction in induction costs. – Please see additional results in final subsection of results (lines 310-351), and expanded discussion detailing the cost saving from applying the oleic acid inducible irreversible switch in lines 433-439.
4. TYPOGRAPHICAL
- Ln226: Fig.4(c) -> Fig. 3(c)
 - Ln229: Fig.4(b) and (c) -> Fig. 3(b) and (c)
 - Ln233: feed-batch -> fed-batch
 - Thank you for bringing these three typos to our attention. They have now been corrected (highlighted in red text in the manuscript).

Relevant references

1. Janßen, H. J. & Steinbüchel, A. Fatty acid synthesis in *Escherichia coli* and its applications towards the production of fatty acid based biofuels. *Biotechnol. Biofuels* **7**, 7 (2014).
2. Matsuoka, H., Hirooka, K. & Fujita, Y. Organization and function of the YsiA regulon of *Bacillus subtilis* involved in fatty acid degradation. *J. Biol. Chem.* **282**, 5180–5194 (2007).
3. Banchio, C. & Gramajo, H. C. Medium- and long-chain fatty acid uptake and utilization by *Streptomyces coelicolor* A3(2): First characterization of a Gram-positive bacterial system. *Microbiology* **143**, 2439–2447 (1997).
4. Irzik, K. *et al.* Acyl-CoA sensing by FasR to adjust fatty acid synthesis in *Corynebacterium glutamicum*. *J. Biotechnol.* **192**, 96–101 (2014).
5. Klug, L. & Daum, G. Yeast lipid metabolism at a glance. *FEMS Yeast Res.* **14**, 369–388 (2014).
6. Wen, Z., Zhang, S., Odoh, C. K., Jin, M. & Zhao, Z. K. *Rhodospiridium toruloides* - A potential red yeast chassis for lipids and beyond. *FEMS Yeast Res.* **20**, 1–12 (2020).
7. Hynes, M. J., Murray, S. L., Duncan, A., Khew, G. S. & Davis, M. A. Regulatory genes controlling fatty acid catabolism and peroxisomal functions in the filamentous fungus *Aspergillus nidulans*. *Eukaryot. Cell* **5**, 794–805 (2006).
8. Su, X. & Abumrad, N. A. Cellular Fatty Acid Uptake: A Pathway Under Construction. *Trends Endocrinol. Metab.* **20**, 72–77 (2009).
9. Dirusso, C. C., Tsvetnitsky, V., Højrup, P. & Knudsen, J. Fatty Acyl-CoA Binding Domain of the Transcription Factor FadR. **273**, 33652–33659 (1998).
10. Mannan, A. A., Liu, D., Zhang, F. & Oyarzún, D. A. Fundamental design principles for transcription-factor-based metabolite biosensors. *ACS Synth. Biol.* **6**, 1851–1859 (2017).
11. Gardner, T. S., Cantor, C. R. & Collins, J. J. Construction of a genetic toggle switch in *Escherichia coli*. *Nature* **403**, 339–342 (2000).

Reviewers' Comments:

Reviewer #1:

Remarks to the Author:

Thanks for addressing my comments. The paper reads well

Reviewer #2:

Remarks to the Author:

My comments have all been addressed. The additional comparison to the genetic toggle switch is nice work and really highlights the advantages of this approach.

Overall, this is very nice paper providing both a clear and accessible description of the research combined with detailed analysis in the supplementary information. Hopefully you can convince an industrial partner to implement the system!

I noticed one typo:

line 48: "allow" -> "allows"

Response to Reviewers

Designing an irreversible metabolic switch for scalable induction of microbial chemical production
Ahmad A. Mannan and Declan G. Bates

We thank both reviewers for their very positive remarks to our responses and for acknowledging all their comments have been addressed.

Reviewer #1

Thanks for addressing my comments. The paper reads well

We are glad they feel the paper reads well. There were no additional points raised that needed to be addressed.

Reviewer #2

My comments have all been addressed. The additional comparison to the genetic toggle switch is nice work and really highlights the advantages of this approach.

Overall, this is very nice paper providing both a clear and accessible description of the research combined with detailed analysis in the supplementary information. Hopefully you can convince an industrial partner to implement the system!

I noticed one typo:

line 48: "allow" -> "allows"

We thank the reviewer for their very encouraging comments and for spotting the typo. We have changed the text as suggested, which is highlighted in red text on page 2 of the revised manuscript.